# Antibiotic Resistance Profile of *Salmonella* sp. Isolates from Commercial Laying Hen Farms in Central-Western Brazil

**DOI:** 10.3390/microorganisms12040669

**Published:** 2024-03-27

**Authors:** Dunya Mara Cardoso Moraes, Ana Maria De Souza Almeida, Maria Auxiliadora Andrade, Eduardo de Paula Nascente, Sabrina Castilho Duarte, Iolanda Aparecida Nunes, Valéria De Sá Jayme, Cíntia Minafra

**Affiliations:** 1Department of Veterinary Medicine, Federal University of Goiás, Goiania 74605-080, Goiás, Brazil; dunyamaramoraes@gmail.com (D.M.C.M.); ana_almeida@ufg.br (A.M.D.S.A.); maa@ufg.br (M.A.A.); eduardodepaula100@gmail.com (E.d.P.N.); iolanda_nunes@ufg.br (I.A.N.); sa.jayme@ufg.br (V.D.S.J.); 2Brazilian Agricultural Research Corporation (Embrapa), Brasília 70770-901, Brazil; sabrina.duarte@embrapa.br; 3Center for Food Research, Federal University of Goiás, Goiania 74660-970, Goiás, Brazil

**Keywords:** *bla*TEM, *intl*1, multidrug resistance, *sul*1

## Abstract

Microbial resistance to antibiotics poses a significant threat to both human and animal health, necessitating international efforts to mitigate this issue. This study aimed to assess the resistance profiles of *Salmonella* sp. isolates and identify the presence of *intl*1, *sul*1, and *bla*TEM resistance genes within antigenically characterized isolates, including Agona, Livingstone, Cerro, Schwarzengrund, *Salmonella* enterica subsp. enterica serotype O:4.5, Anatum, Enteritidis, Johannesburg, Corvallis, and Senftenberg. These isolates underwent susceptibility testing against 14 antibiotics. The highest resistance percentages were noted for sulfamethoxazole (91%), sulfonamides (51%), and ceftiofur (28.9%), while no resistance was observed for ciprofloxacin. *Salmonella* Johannesburg and *Salmonella* Corvallis showed resistance to one antibiotic, whereas other serovars were resistant to at least two. *Salmonella* Schwarzengrund exhibited resistance to 13 antibiotics. The *intl*1 gene was detected in six out of the ten serovars, and the *sul*1 gene in three, always co-occurring with *intl*1. The *bla*TEM gene was not identified. Our findings highlight the risk posed by the detected multiple resistances and genes to animal, human, and environmental health. The multidrug resistance, especially to third-generation cephalosporins and fluoroquinolones, highlights the need for stringent monitoring of *Salmonella* in laying hens. The potential of the environment, humans, eggs, and their products to act as vectors for antibiotic resistance represents a significant concern for One Health.

## 1. Introduction

Salmonellosis is a globally widespread foodborne illness [1] with significant implications for public health worldwide, as humans often contract multidrug-resistant (MDR) *Salmonella* infections through the consumption of contaminated animal products [2,3].

MDR *Salmonella* is characterized by its resistance to conventional first-line antibiotics such as ampicillin, chloramphenicol, and trimethoprim-sulfamethoxazole [4,5,6]. The global rise in the isolation of *Salmonella* serotypes resistant to one or more antibiotics [7] is largely attributed to the inappropriate use, overuse, and easy accessibility of antibiotics across various countries [2]. In South Africa, *Salmonella* has been frequently isolated from animal feeds, livestock, and their environments [8]. The European Union has reported MDR *Salmonella* in poultry meat samples [9]. In New South Wales and South Australia, *Salmonella* spp. isolated from commercial caged layer flocks exhibited the highest resistance to amoxicillin and ampicillin, followed by tetracycline, cephalothin, and trimethoprim. In Brazil, a significant portion (80.9%) of *Salmonella* isolates from different stages of swine production were deemed multidrug-resistant (resistant to ≥3 antibiotic classes), showing the greatest resistance to streptomycin (90.5%), tetracycline (88.1%), ampicillin (81.0%), chloramphenicol (71.4%), and ciprofloxacin (50.0%) [10]. In China, *Salmonella* strains isolated from broilers and poultry workers displayed MDR phenotypes, resisting up to 16 commonly used antibiotics such as ampicillin, streptomycin, kanamycin, amikacin, tetracycline, nalidixic acid, ciprofloxacin, trimethoprim/sulfamethoxazole, and chloramphenicol [11].

A global review on antibiotic resistance in *Salmonella* sp. isolated from poultry [12] revealed that the median prevalence rates in broilers, raw chicken meat, eggs, and laying hens were 40.5%, 30%, and 40%, respectively. The study highlighted that the highest antibiotic resistance levels within the poultry production chain were observed for nalidixic acid and ampicillin (Figure 1). MDR *Salmonella* presence has also been reported in the layer farm environment, suggesting public health risks associated with egg consumption [13,14] and representing a significant risk factor for salmonellosis [15]. Another study focusing on egg farm environments in Western Australia reported a high incidence of *Salmonella* [14].

While antibiotic resistance can occur naturally, the emergence of most resistant organisms is attributed to the acquisition of resistance genes through mutations or the transfer of genetic material, followed by selective pressure [16,17,18,19,20,21].

Food should contribute positively to human health, and over time, the concept of functional foods has evolved, emphasizing foods that lower the risk of various health issues [22,23]. However, the food chain is also a critical vector for the spread of antibiotic resistance from animals to humans. The rise of antibiotic resistance in non-typhoidal *Salmonella* strains poses a significant public health risk [24,25,26], underscoring the importance of monitoring resistance profiles in *Salmonella* Typhimurium and *Salmonella* Enteritidis. This not only aids in disease management but also helps track the development of potential multidrug-resistant strains. Resistance mechanisms can be shared among bacteria within the same genus or across different bacterial groups [27,28].

Genotypic characterization techniques such as quantitative PCR (qPCR) offer a more precise identification of *Salmonella* serovars by detecting specific DNA sequences. These methods surpass phenotypic characterization in discriminatory capability [29,30,31]. A characterization of *Salmonella* incorporating multi-locus sequence typing (MLST), genotype antibiotic resistance (AMR) profiles, plasmid incompatibility types, phylogeny, and serotype can elucidate the clonal populations. The diversity of AMR genes in *Salmonella* from various sources suggests genome alterations due to the acquisition of new AMR genes through mutations, plasmid transfers, or existing natural resistance genes [32].

Genotypic methods that analyze chromosomal or plasmid DNA enable the identification of bacteria through unique or species-specific genes, as well as the detection of particular genes [33,34]. These techniques can help to understand the development and transferability of mobile resistance-encoding elements in bacterial pathogens, driven by interactions between specific chromosomal fragments and plasmids [35]. As the prevalence of transferable virulence and antibiotic resistance increases, the role of antibiotic-resistant plasmids in *Salmonella* becomes a growing concern for both animal and public health [36].

This study aimed to determine the antibiotic resistance profiles of *Salmonella* serovars from commercial egg and laying hen farms in the Center-West region of Brazil and to identify the presence of *intl*1, *sul*1, and *bla*TEM resistance genes. This research represents the first phenotypic and genotypic mapping of *Salmonella* isolates conducted over a year in an egg and by-product production region in Brazil.

## 2. Materials and Methods

For this study, *Salmonella* serovars preserved in the bacteriology laboratory were utilized. The research involved two main assessments: (1) antibiotic susceptibility profiling, in which the *Salmonella* serovars were subjected to various classes of antibiotics; and (2) gene detection, employing qPCR assays to identify antibiotic resistance genes across all *Salmonella* serovars.

### 2.1. Study Location

The project was carried out in Goiânia, GO, Brazil (−16.67926° N −49.25629° W).

### 2.2. Sampling

The study analyzed 45 *Salmonella* isolates retrieved from samples collected over a year from commercial egg and layer farms. Laboratory identification of the *Salmonella* genus was conducted through biochemical testing and further verified using a polyvalent antiserum “O” agglutination test for *Salmonella* (Probac, São Paulo, Brazil). Serotyping of the isolates was performed at the National Reference Laboratory for Enteric Diseases at the Oswaldo Cruz Institute—FIOCRUZ (NRL).

These serovars, preserved on nutrient agar, were contributed to this research by the Laboratory of Bacteriology at the Faculty of Veterinary Medicine, Federal University of Goiás. The serovars used in this study were Agona (14/45), Livingstone (8/45), Cerro (6/45), Schwarzengrund (5/45), Enterica O:4.5 (4/45), Anatum (3/45), Enteritidis (2/45), Johannesburg (1/45), Corvallis (1/45), and Senftenberg (1/45).

### 2.3. Antibiotic Susceptibility Profiling

A total of 45 *Salmonella* isolates cultured on nutrient agar were further plated on bright green agar and incubated at 37 °C for 24 h. Following incubation, colony-forming units exhibiting typical *Salmonella* morphology traits were identified and selected. Two colony-forming units from each plate were then transferred to tubes containing Triple Sugar Iron (TSI) agar and incubated at 37 °C for an additional 24 h.

The antibiotic susceptibility profile of 45 isolates was assessed using the disk diffusion method, adhering to the performance standards for antibiotic susceptibility testing [37]. The antibiotics tested, along with their respective classes and concentrations, are presented in Table 1.

Isolates confirmed to display *Salmonella* characteristics were then transferred to 5 mL of Casoy broth and incubated until a turbidity of 0.5 on the McFarland scale was achieved. Subsequently, a swab immersed in the broth was drained by pressing it against the walls of the tube to eliminate excess fluid, and then streaked across a Mueller-Hinton agar plate to ensure a homogeneous distribution of the inoculum. The inoculated plates were left undisturbed for 15 min to allow the inoculum to be adequately absorbed by the agar. Sterilized tweezers were then used to carefully place antibiotic discs onto the inoculated surface.

Each disc was gently pressed down to enhance adherence and spaced approximately 3 cm apart. The plates were then incubated upside down at 37 °C for 24 h. Post-incubation, the zones of inhibition were measured with a ruler, and the results were interpreted based on standardized tables that account for the concentration of the discs used. Antibiotic resistance prevalence data were summarized using the median and interquartile range (IQR) across studies, with the analysis conducted using EpiInfo2000 version 3.3.1 (CDC—Atlanta, GA, USA, 2005). The resistance frequency to each specific antibiotic was tabulated, and isolates were classified as MDR if they exhibited resistance to three or more classes of antibiotics.

### 2.4. Gene Detection

Total DNA extraction was conducted using the boiling lysis method [38]. For each sample, 400 μL was transferred to a 1.5-mL DNA- and RNA-free polypropylene tube (Axygen, Union City, CA, USA) and centrifuged at 2000× *g* for 4 min. The supernatant was discarded, and the pellet was resuspended in 1 mL TE buffer (100 mL Tris/HCl 1 m, 20 μL EDTA 0.5 m, and 9.880 μL H_2_O), vortexed for 10 s, and centrifuged again at 2000× *g* for 8 min. Once again, the supernatant was discarded, and the pellet was resuspended in 100 μL TE buffer, vortexed for 10 s, and placed on a hot plate at 95 °C for 20 min. The lysate was then aliquoted and stored in a freezer at −20 °C for subsequent analysis.

qPCR assays to detect *intl*1, *sul*1, and *bla*TEM genes in *Salmonella* sp. were conducted following the protocol described by Bugarel et al. [39]. The reaction mixture for qPCR, using the TaqMan system, comprised 20 μL per sample: 4.6 μL of milli-Q water, 10 μL of master mix (1×), 2 μL of IPC mix (10×), 0.4 μL of IPC DNA (50×), and 1 μL of oligonucleotide primers (30 mM concentration) and probe (10 mM concentration), with 2 μL of the DNA extract added. An internal control was included in one of the wells of a 96-well plate using IPC DNA with and without IPC blocker reagent (Life^®^, Thermo Fischer Scientific, Waltham, MA, USA) as the negative control. Samples were tested for presence/absence in the StepOnePlus™ qPCR system (Applied Biosystems, Waltham, MA, USA) under the following PCR conditions: an initial hold at 60 °C for 30 s, a denaturation step at 95 °C for 10 min, followed by 40 cycles of 95 °C for 15 s and 60 °C for 1 min and 30 s for the extension step.

The real-time PCR detection of genes employed the TaqMan system with specific oligonucleotide primers for *intl*1 (f 5′-TGGGCAGCAGCGAAGTC-3′, r 5′-TGCGTGGAGACCGAAACC-3′, probe FAM-AGGCATTTCTGTCCTGGCTGGCG-BHQ), *sul*1 (f 5′-TCCTGACCCTGCGCTCTATC-3′, r 5′-TGCGCTGAGTGCATAACCA-3′, probe ROX-ATTGCTGAGGCGGACTGCAGGC-BHQ), and *bla*TEM (f 5′-CTGGATCTCAACAGCGG-3′, r 5′-CAACACGGGATAATACCGC, probe FAM-AGATCCTTGAGAGTTTTCGCCCCG-BHQ).

Adjustments to the apparatus were made to accommodate available filters, replacing ROX with FAM and BHQ with TAMRA as needed.

Statistical analysis to determine significant differences between gene frequencies utilized Pearson’s chi-squared test or Fisher’s exact test, as appropriate, with *p* ≤ 0.05 denoting statistical significance. All statistical analyses were executed using EpiInfo2000 version 3.3.1 (CDC—Atlanta, 2005).

## 3. Results

The highest resistance percentages observed were to sulfamethoxazole (91%) followed by sulfonamides (51%) and ceftiofur (28.9%). Only ciprofloxacin showed no resistance among the tested serovars (Table 2). Resistance to just one antibiotic, sulfamethoxazole, was noted exclusively in *Salmonella* Johannesburg and *Salmonella* Corvallis, whereas all other examined serovars displayed resistance to at least two antibiotics (Table 3). *Salmonella* Schwarzengrund showed resistance to thirteen antibiotics, followed by *Salmonella* Enteritidis and *Salmonella* enterica subsp. enterica serotype O:4.5, each resistant to five antibiotics. *Salmonella* Agona exhibited resistance to four antibiotics, while both *Salmonella* Livingstone and *Salmonella* Cerro were resistant to three. *Salmonella* Senftenberg and another unidentified *Salmonella* serovar showed resistance to two antibiotics. Serovars such as Agona, *Salmonella* enterica subsp. enterica serotype O:4.5, and Schwarzengrund were identified as MDR, demonstrating resistance to three or more classes of antibiotics (Figure 2).

The presence of resistance genes *sul*1 and *intl*1 was confirmed (Table 4). Class 1 integrons were found in serovars Cerro, Johannesburg, and *Salmonella* enterica subsp. enterica O:4.5 and was associated with the *sul*1 gene in serovars Agona, Livingstone, and Schwarzengrund.

Ampicillin resistance was detected in 1/5 (20%) of the Schwarzengrund isolates. Resistance to amoxicillin-clavulanic acid was observed in 1/14 (7.1%) of Agona isolates and 1/5 (20%) of Schwarzengrund isolates. The *bla*TEM gene, however, was not detected in any of the tested isolates.

## 4. Discussion

The indiscriminate use of antibiotics in human and animal healthcare, as well as in food production, coupled with their subsequent release into the environment, contribute to the emergence of antibiotic-resistant *Salmonella* strains [40]. Understanding the link between antibiotic resistance and the presence of resistance genes in *Salmonella* strains is crucial and warrants continuous investigation [17,19,22,26,41]. Contaminated poultry products represent the primary transmission vector for *Salmonella*. Therefore, monitoring antibiotic resistance in *Salmonella* originating from poultry is essential [42]. The assessment of both phenotypic and genotypic resistance in *Salmonella* isolated from commercial eggs in Brazil reveals a high prevalence within the egg production sector, highlighting the need for effective management and biosecurity measures. Research focusing on *Salmonella* in eggs has become crucial in preventing foodborne diseases, with various serovars being isolated annually from both eggs and environmental samples such as laying hen farms [43].

Tajbakhsh et al. [44] reported a human *Salmonella* isolate in Iran showing resistance to sulfamethoxazole (30%), trimethoprim-sulfamethoxazole (15%), and ampicillin (14%), which differs from the findings of this study possibly due to variations in the sources of *Salmonella* isolation. This discrepancy underscores the presence of resistant bacteria across food, animals, and humans. Lozano-Villegas et al. [45] suggested that the variation in *Salmonella* strains isolated from broilers and humans could be attributed to differences in pathogenic potential, influenced by the type and number of detected virulence genes, including those linked to antibiotic resistance.

Hu et al. [46] observed that among 143 *Salmonella* isolates from eggs and chicken sources across different countries, there was a high prevalence of genes conferring resistance to aminoglycosides (70.63%), tetracyclines (26.57%), and β-lactamases (15.38%), indicating higher resistance levels than those reported in this study. Barlow et al. [24] highlighted an annual 30% increase in the likelihood of infections caused by bacteria resistant to quinolones and trimethoprim-sulfamethoxazole, emphasizing the public health significance of salmonellosis and the critical need for judicious antibiotic use in both human and veterinary medicine.

In contrast, Galdino et al. [47], who analyzed the susceptibility profiles of 18 avian *Salmonella* sp. isolates from poultry fecal samples, found no resistance to neomycin or tetracycline. This is in contrast to our findings, which indicate 75.6% intermediate resistance to neomycin and 15.5% resistance to tetracycline. Additionally, the resistance to sulfonamides was reported at 51% in this study, whereas the aforementioned researchers reported a significantly lower rate of 11%. This variance could be attributed to the different sources of *Salmonella* isolation, with [47] focusing on environmental samples rather than eggs.

A study on food poisoning within the European Union focusing on the susceptibility profile of *Salmonella* sp. indicated resistance levels of 21–35% to ampicillin, 36–52% to sulfonamides, and 38–59% to tetracycline. These figures for ampicillin and tetracycline are considerably higher than those observed in this study, which were 4.4% and 2.2%, respectively. Moreover, *Salmonella* strains isolated from chicken sources have exhibited high levels of MDR. Among these, ampicillin resistance was most prevalent, with resistance genes such as *bla*TEM, *sul*1, and *tet*A being detected and linked to MDR [48]. MDR *Salmonella* isolates from fecal swabs showed a similar trend, with 100% harboring the bla gene and 77% possessing the *tet*A extended-spectrum beta-lactamase (ESBL) gene [49].

However, the resistance to sulfonamides in this study was found to be similar (51%). In terms of ciprofloxacin resistance, no resistant isolate was identified, and a 4.4% resistance rate was observed for enrofloxacin. These findings are closely aligned with those reported in Italy and Denmark, where resistance rates of 1% to ciprofloxacin and 0.6% to enrofloxacin were recorded [50].

In a distinct study conducted in Goiás, Brazil, among 32 *Salmonella* isolates obtained from drag swabs collected from 23 chicken farms, 29 (90.6%) exhibited resistance to ceftiofur [49]. Nonetheless, the level of resistance to ceftiofur noted in this investigation was 28.9%, and intermediate resistance levels to ciprofloxacin and enrofloxacin were observed. Given that fluoroquinolones and third- and fourth-generation cephalosporins are classified by the World Health Organization (WHO) as “critically important” antibiotics—used in treating serious human infections and diseases caused by organisms transmitted from non-human sources or capable of acquiring resistance genes—the resistance levels noted in this study, particularly to ceftiofur and intermediate resistance to ciprofloxacin and enrofloxacin, raise concerns [51].

In this study, Enteritidis isolates showed no resistance to ampicillin, doxycycline, or tetracycline, contrasting with findings by Lu et al. [31] who observed resistance to the above three antibiotics in avian *Salmonella* Enteritidis samples collected from fecal samples (chicken hatcheries, farms, and slaughterhouses) in China. Similarly, Ribeiro et al. [52] reported a 67.1% resistance rate to tetracycline and a 22.8% resistance rate to gentamicin in *Salmonella* Enteritidis isolates of avian origin (clinical and environmental poultry samples) from southern Brazil. These data differ from the findings in this study, where no *Salmonella* Enteritidis isolate showed resistance to tetracycline, and 50% exhibited resistance to gentamicin. The discrepancy, especially in tetracycline resistance, may be attributed to the ban on tetracycline use in veterinary medicine for production animals, which likely contributed to the reduced prevalence of the *tet*A gene [53,54,55]. Furthermore, *Salmonella* Enteritidis isolates from eggs demonstrated high resistance gene prevalence rates, including the *spv*C gene, primarily located on the *Salmonella* virulence plasmid, detected in 50.8% of *Salmonella* Enteritidis samples, with 27% of this serovar being MDR [56].

Cui et al. [57], in their analysis conducted in China, found that among 25 Agona isolates, 40% were resistant to ampicillin, 32% to trimethoprim-sulfamethoxazole, and 28% to gentamicin. These results are akin to resistance observed in other β-lactams tested in *Salmonella* isolates from fecal swabs in Brazil, including a 34% resistance rate to amoxicillin-clavulanic acid. Moreover, 81% of these isolates were also resistant to non-beta-lactam antibiotics such as ciprofloxacin and enrofloxacin, 72% to tetracycline, and 97% to trimethoprim-sulfamethoxazole [49]. This variation in resistance among *Salmonella* isolates from different regions may be due to regional peculiarities in antibiotic usage [58].

According to Hsu et al. [59], the detection of the *intl*1 gene is closely related to the presence of *Salmonella* Genomic Island 1 (SGI-1) and indicates the presence of a class 1 integron. This structure houses a cassette of genes that promote antibiotic resistance, including the *sul*1 gene, which is responsible for resistance to sulfonamides.

The class 1 integron is commonly found in *Salmonella* Typhimurium [60,61,62,63,64], but in this study, it was identified solely in serovars Cerro, *Salmonella* enterica subsp. enterica O: 4.5, and Johannesburg and was associated with the *sul*1 gene in the serovars Agona, Livingstone, and Schwarzengrund (Table 3). It is likely that these serovars, sharing the same habitat, have acquired resistance genes on plasmids from other bacteria. This hypothesis is supported by Bloomfield et al. [65], who described plasmids as the primary vehicles for the acquisition of resistance genes in the genus *Salmonella*.

The detection and frequency of the *intl*1 gene in isolates of serovars Cerro, Schwarzengrund, *Salmonella* enterica subsp. enterica O: 4.5, and Johannesburg, all from the same environment, echo the findings of Alam et al. [66], who reported the *intl*1 presence across various serovars, likely acquired through horizontal gene transfer.

Table 3 reveals the absence of the *intl*1 gene in serovars Anatum, Enteritidis, Corvallis, and Senftenberg. Moreover, the *sul*1 gene was not detected in these serovars, nor in isolates from Johannesburg, Cerro, or *Salmonella* enterica subsp. enterica O:4.5, being found exclusively alongside the *intl*1 gene.

Hsu et al. [59] noted that multidrug resistance is closely related to the presence of class 1 integrons. In this study, serovar Corvallis did not exhibit the *intl*1 gene or MDR, while serovars Anatum and Senftenberg showed resistance to two antibiotics, and Enteritidis to five, without the detection of the *intl*1 gene among them. It is plausible that the observed MDR in serovars Anatum, Senftenberg, and Enteritidis might be associated with class 2 or 3 integrons, which, as Yang et al. [67] suggest, can confer resistance to numerous antibiotics. Firoozeh et al. [68] and Ramatla et al. [69] identified class 1 integrons predominantly in plasmids and transposons, whereas class 2 and 3 integrons are usually found in transposons, with class 1 being more prevalent than class 2. Notably, class 3 integrons have not been reported in *Salmonella* species, a finding consistent with this research.

Table 3 indicates that all three Livingstone isolates exhibited phenotypic resistance to sulfonamides and harbored the *sul*1 gene. However, among the sulfonamide-resistant isolates from Cerro, Schwarzengrund, and Salmonella enterica subsp. enterica O:4.5, the *sul*1 gene was detected in only one Cerro isolate, two Schwarzengrund isolates, and one *Salmonella* enterica subsp. enterica O:4.5 isolate. This observation aligns with Wannaprasat et al. [70], who identified the *sul*3 gene, which confers sulfonamide resistance and is associated with class 1 integrons, in the absence of *sul*1.

Our ampicillin data differ from those reported by Hawkey et al. [58], who detected the *bla*TEM gene in 18% of human-derived *Salmonella* isolates resistant to this antibiotic, our study found no *bla*TEM gene presence. Similarly, *Salmonella* isolates resistant to cefotaxime from broiler litter samples were *bla*TEM-negative [71]. Nonetheless, the *bla*TEM gene was present in 79.2% and the t*et*A gene in 75% of *Salmonella* isolates from sulfonamide-treated diarrheic calves [72].

The results from this study, focusing on *Salmonella* isolates from commercial egg and laying hen farms, reveal a resistance pattern to sulfamethoxazole, sulfonamides, and ceftiofur. This resistance profile may stem from continuous exposure to these antibiotics and subsequent selective pressure, leading to the transfer of resistance genes such as *intl*1, *sul*1, and *bla*TEM between strains. Understanding antibiotic resistance profiles in *Salmonella* from chickens, laying hens, and eggs is crucial for reassessing regulations and formulating more effective resistance prevention strategies, as MDR *Salmonella* with zoonotic potential can transfer resistance genes from animals to humans [40].

In layer farms and commercial eggs from Central-Western Brazil, the occurrence of MDR *Salmonella* serovars such as Agona, *Salmonella* enterica subspecies enterica O:4.5, and Schwarzengrund may pose a public health risk [42,73]. Our study contributes valuable data on resistance profiles and MDR *Salmonella* in chickens, identifying multiple resistances to third-generation cephalosporins and fluoroquinolones, deemed “critically important antibiotics” (CIA).

The National Plan for Residue and Contaminant Control [74] has identified antibiotics such as ciprofloxacin, enrofloxacin, and florfenicol in high concentrations in commercial eggs. Given that the central-western region of Brazil accounts for 13.06% of the country’s chick production and is a major exporter of fresh eggs [75], this research offers significant insights into the potential risks for layer farms and commercial egg production. Therefore, the phenotypic and genotypic resistance of *Salmonella* to various antibiotics demands attention concerning environmental, human, and animal health.

## 5. Conclusions

*Salmonella* isolates exhibited elevated resistance rates to sulfamethoxazole, sulfonamides, and ceftiofur, while no resistance to ciprofloxacin was detected. *Salmonella* Johannesburg and *Salmonella* Corvallis were resistant to a single antibiotic, sulfamethoxazole. In contrast, the other serovars demonstrated resistance to two or more antibiotics, with *Salmonella* Schwarzengrund showing resistance to thirteen antibiotics. The detection of the *spv*C gene in *Salmonella* Enteritidis highlights the urgency of controlling this serovar due to its virulence-associated structures. The identification of the SGI-1 genomic island, indicated by the presence of the *intl*1 gene in six out of the ten serovars studied and the *sul*1 gene in three, suggests that *Salmonella* strains in the commercial egg production sector harbor genes facilitating host invasion and survival. The monitoring and control of *Salmonella* sp., through the identification of virulence and antibiotic resistance genes, are critical for the safety of commercial egg production. Government bodies, researchers, and poultry producers have a responsibility to address antibiotic resistance through judicious antibiotic use, active surveillance of MDR strains, and exploration of alternative disease control and prevention measures, reflecting the critical importance of these efforts in maintaining food chain integrity. Additionally, the mapping of phenotypic and genotypic antibiotic resistance in *Salmonella* serovars aids in informing governmental strategies to ensure the effectiveness of surveillance systems. Given the classification of *Salmonella* by the World Health Organization (WHO) as a high-priority pathogen in the context of antibiotic resistance, this issue constitutes a global challenge with wide-ranging implications across the One Health spectrum.

## Figures and Tables

**Figure 1 microorganisms-12-00669-f001:**
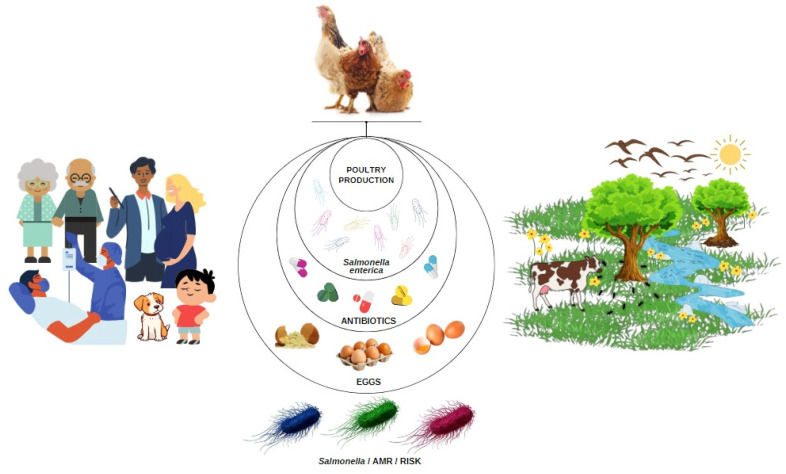
Representation of interactions between layer production system, presence of antibiotic-resistant *Salmonella enterica*, environment, consumers, and consumption of eggs and by-products as a scenario of the risk of antibiotic resistance (AMR) for One Health.

**Figure 2 microorganisms-12-00669-f002:**
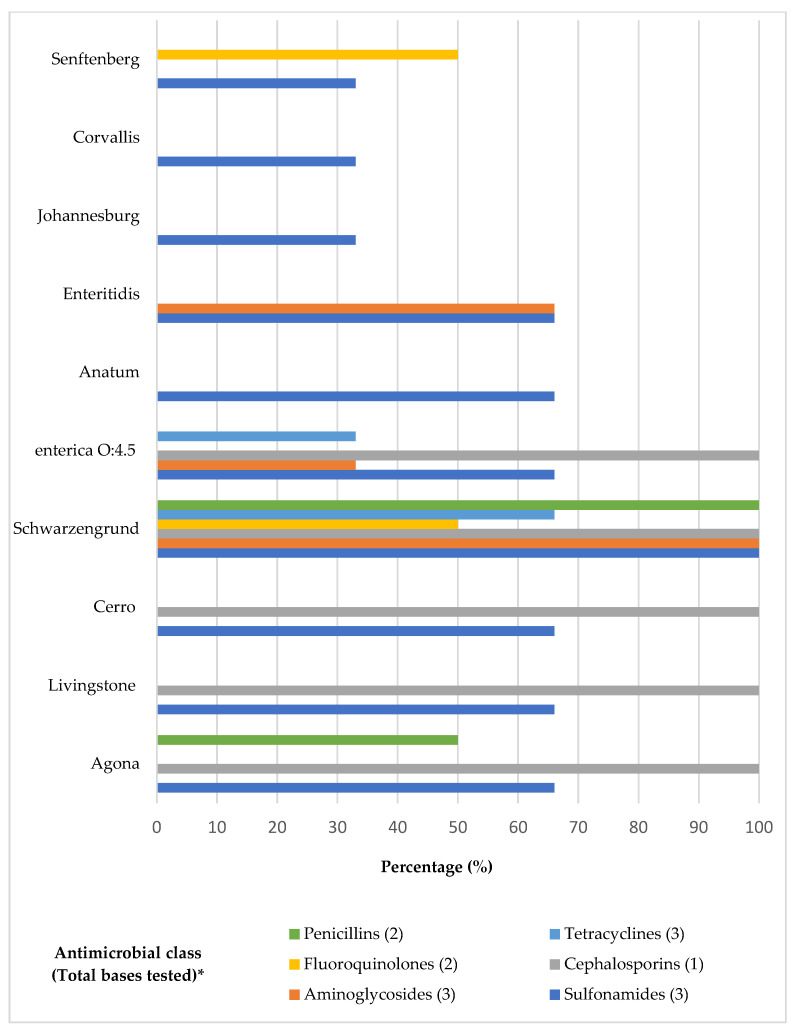
Distribution of *Salmonella* serovars resistant to antibiotic classes. * Tested bases of each antibiotic class: Penicillins (Amoxicillin clavulanic acid, Ampicillin); Fluoroquinolones (Enrofloxacin; Ciprofloxacin); Aminoglycosides (Gentamicin, Neomycin, Apramycin); Tetracyclines (Tetracycline, Oxytetracycline, Doxycycline); Cephalosporins (Ceftiofur); Sulfonamides (Sulfamethoxazole, Trimethoprim-sulfamethoxazole, Sulfonamides).

**Table 1 microorganisms-12-00669-t001:** Antibiotic classes, antibiotics tested, and their respective concentrations used in antibiotic susceptibility profiling.

Antibiotic Class	Antibiotic Tested	Concentration
Cephalosporins	Ceftiofur	30 μg
Fluoroquinolones	Enrofloxacin	5 μg
	Ciprofloxacin	10 μg
Penicillins	Amoxicillin + Clavulanic acid	3 μg
	Ampicillin	20 μg
Tetracyclines	Doxycycline	30 μg
	Tetracycline	30 μg
	Oxytetracycline	30 μg
Aminoglycosides	Gentamicin	10 μg
	Neomycin	30 μg
	Apramycin	15 μg
Sulfonamides	Trimethoprim + sulfamethoxazole	25 μg
	Sulfamethoxazole	25 μg
	Sulfonamides	300 μg

**Table 2 microorganisms-12-00669-t002:** Antibiotic susceptibility profile of 45 *Salmonella* isolates.

Antibiotic	Susceptibility Profile		
	Resistance (%)	Intermediate Resistance (%)	Sensitivity (%)
Sulfamethoxazole	91	6.7	2.3
Sulfonamides	51	9	40
Ceftiofur	28.9	42.2	28.9
Apramycin	6.7	48.9	44.4
Amoxicillin/clavulanic acid	4.4	20	75.5
Enrofloxacin	4.4	6.7	88.9
Gentamicin	4.4	-	95.6
Neomycin	4.4	75.6	20
Doxycycline	4.4	60	35.5
Tetracycline	2.2	15.5	82.3
Ampicillin	2.2	2.2	95.6
Trimethoprim-sulfamethoxazole	2.2	-	97.8
Oxytetracycline	2.2	8.9	88.9
Ciprofloxacin	-	11.1	88.9

**Table 3 microorganisms-12-00669-t003:** Antibiotic resistance of *Salmonella* sp. serovars.

Serovar (n)	Agona (14)	Livingstone (8)	Cerro (6)	Schwarzengrund (5)	Enterica O:4.5 (4)	Anatum (3)	Enteritidis (2)	Johannesburg (1)	Corvallis (1)	Senftenberg (1)	Total (45)
SMT	11/14 DFPA	8/8 DFPA	5/6 DFPA	5/5 DFPA	4/4 DFPA	3/3 DFPA	2/2 DFPA	1/1 DFPA	1/1 DFPA	1/1 DFPA	41
STX	-	-	-	1/5 DFPA	-	-	-	-	-	-	1
ENR	-	-	-	1/5 DFPA	-	-	-	-	-	1/1 DFPA	2
TET	-	-	-	1/5 DFPA	-	-	-	-	-	-	1
SUL	6/14 DFPA	3/8 DFPA	2/6 DFPA	4/5 DFPA	4/4 DFPA	2/3 DFPA	2/2 DFPA	-	-	-	23
CIP	-	-	-	-	-	-	-	-	-	-	-
AMC	1/14 DFPA	-	-	1/5 DFPA	-	-	-	-	-	-	2
AMP	-	-	-	1/5 DFPA	-	-	-	-	-	-	1
CEF	4/14 DFPA	2/8 DFPA	1/6 DFPA	3/5 DFPA	3/4 DFPA	-	-	-	-	-	13
GEN	-	-	-	1/5 DFPA	-	-	1/2 DFPA	-	-	-	2
OXI	-	-	-	1/5 DFPA	-	-	-	-	-	-	1
NEO	-	-	-	1/5 DFPA	-	-	1/2 DFPA	-	-	-	2
DOX	-	-	-	1/5 DFPA	1/4 DFPA	-	-	-	-	-	2
APR	-	-	-	1/5 DFPA	1/4 DFPA	-	1/2 DFPA	-	-	-	3

Legends: SMT—sulfamethoxazole (25 μg); STX—trimethoprim-sulfamethoxazole (25 μg); ENR—enrofloxacin (5 μg); TET tetracycline (30 μg); SUL—sulfonamides (300 μg); CIP—ciprofloxacin (10 μg); AMC—amoxicillin clavulanic acid (3 μg); AMP—ampicillin (20 μg); CEF—ceftiofur (30 μg); GEN—gentamicin (30 μg); OXI—oxytetracycline (30 μg); NEO—neomycin (30 μg); DOX—doxycycline (30 μg); APR—apramycin (15 μg).

**Table 4 microorganisms-12-00669-t004:** Resistance to ampicillin (AMP), amoxicillin with clavulanic acid (AMC), and sulfonamides (SUL) and qPCR for *intl*1, *sul*1, and *bla*TEM resistance genes in different serovars.

*Salmonella* Serovar	n	AMP	AMC	SUL	Resistance Gene		
					*intl*1	*sul*1	*bla*TEM
Agona	14	-	1/14 DFPA	6/14 DFPA	1/6 DFPA	1/6 DFPA	-
Livingstone	8	-	-	3/8 DFPA	3/8 DFPA	3/8 DFPA	-
Cerro	6	-	-	2/6 DFPA	1/6 DFPA	-	-
Schwarzengrund	5	1/5 DFPA	1/5 DFPA	4/5 DFPA	2/5 DFPA	2/5 DFPA	-
enterica O:4.5	4	-	-	4/4 DFPA	1/4 DFPA	-	-
Anatum	3	-	-	2/3 DFPA	-	-	-
Enteritidis	2	-	-	2/2 DFPA	-	-	-
Johannesburg	1	-	-	-	1/1 DFPA	-	-
Corvallis	1	-	-	-	-	-	-
Senftenberg	1	-	-	-	-	-	-

## Data Availability

The data presented in this study are available upon request from the corresponding author (privacy).

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
