# Peer review of "Antibiotic Resistance Profile of Salmonella sp. Isolates from Commercial Laying Hen Farms in Central-Western Brazil"

_microorganisms, 2024, doi:10.3390/microorganisms12040669_

Round 1
Reviewer 1 Report
Comments and Suggestions for Authors
Some concerns should be addressed as follows:
1. Abstract: It should be revised carefully. Line 15: determinate should be determine and the same in line 26. Significant findings should be highlighted. The conclusion is not clear. Line 19: antibacterial or antimicrobials, please explain.
2. Introduction: the novelty of this study should be explained at the end of the section.
3. Methods: it should be divided into subsections. Line 102-106: please mention the class of the antibiotics used in this study. Please add a scheme for the isolation process and downstream steps.
4. The authors should scrutinize the whole manuscript and rectify some typos.
5. Results: the authors are encouraged to add figures to clearly explain their findings. I did not find any supplementary data to check.
6. Discussion: the mechanism of bacterial resistance of Salmonella sp. to antibiotics should be clarified.
Comments on the Quality of English LanguageThe authors should scrutinize the whole manuscript and rectify some typos.
Author Response
1. Abstract: It should be revised carefully. Line 15: determinate should be determine and the same in line 26. Significant findings should be highlighted. The conclusion is not clear. Line 19: antibacterial or antimicrobials, please explain.
Line 15: I replaced determinate with determine.
Significant findings and the conclusion have been reformulated.
The information was inserted on line 19.
2. Introduction: the novelty of this study should be explained at the end of the section.
The novelty of this study was explained on lines 97-102.
3. Methods: it should be divided into subsections. Line 102-106: please mention the class of the antibiotics used in this study. Please add a scheme for the isolation process and downstream steps.
Methods was divided into subsections.
Line 135-137: The class of the antibiotics used in this study were mentioned.
Line 104-108: We add a scheme with assays
4. The authors should scrutinize the whole manuscript and rectify some typos.
All manuscript was proofread and typographical errors were corrected
5. Results: the authors are encouraged to add figures to clearly explain their findings. I did not find any supplementary data to check.
Line 235: We add a figure to clearly explain their findings
6. Discussion: the mechanism of bacterial resistance of Salmonella sp. to antibiotics should be clarified.
Line 334-338: We add information about mechanism of bacterial resistance of Salmonella
Reviewer 2 Report
Comments and Suggestions for Authors
This study aimed to assess the antimicrobial susceptibility profile of avian Salmonella isolates and also detect the presence of resistance genes intl1, sul1 and blaTEM.
The manuscript describes important laboratory findings, which can have significant clinical relevance. As such, it is very useful and merits publication.
I have added some points below that can be addressed to improve the final version.
-Please provide some details of the history of the isolates.
-The M&M section should be divided in sub-section to allow better flow of reading.
-The Result should be presented visually to enhance the effect of the manuscript and to facilitate readers in understanding it. Please appropriate graphs.
-The Discussion does not cover the important topic of publica health and potential dissemination of the resistance genes. Please add a new paragraph dealing with that aspect. Also, the discussion should explore in greater depth the clinical relevance of the findings.
Overall. The manuscript must be correctly revised and re-evaluated.
Author Response
1. Please provide some details of the history of the isolates.
The history of the isolates was detailed in lines 114 - 124
2. The M&M section should be divided in sub-section to allow better flow of reading.
The M&M section has been divided into subsections
3. The Result should be presented visually to enhance the effect of the manuscript and to facilitate readers in understanding it. Please appropriate graphs.
Line 235: We add a graphic to facilitate understanding.
4. The Discussion does not cover the important topic of publica health and potential dissemination of the resistance genes. Please add a new paragraph dealing with that aspect. Also, the discussion should explore in greater depth the clinical relevance of the findings.
Lines 248-250, 253-255, 334-338, 382-404: We add information about public health potential dissemination of resistance genes and explored the clinical relevance of our findings
Round 2
Reviewer 1 Report
Comments and Suggestions for Authors
The following comments have not been addressed as follows:
1. The authors did not add a scheme to clarify the process of this study.
2. Antimicrobial should be antibacterial in the whole manuscript since we discuss the resistance of salmonella sp.
3. The novelty has not been clarified.
4. Line 137: it is a general sentence about the classification of antibiotics. The authors should specify the class of each antibiotic.
5. Line 427: Supplementary Materials: I did not find SM to check it.
Comments on the Quality of English LanguageThe authors should revise the manuscript carefully.
Author Response
1. The authors did not add a scheme to clarify the process of this study.
Line 66: we created an outline to clarify the process of this study
2. Antimicrobial should be antibacterial in the whole manuscript since we discuss the resistance of Salmonella sp.
We replaced antimicrobial with antibacterial in whole manuscript
3. The novelty has not been clarified.
Line 392-399: we clarified the novelty.
4. Line 137: it is a general sentence about the classification of antibiotics. The authors should specify the class of each antibiotic.
Line 140-141: We created table 1 to make it clear to which class each antibiotic used in our study belongs.
5. Line 427: Supplementary Materials: I did not find SM to check it.
Line 423-430: We have corrected the supplementary materials
After corrections, we submit the manuscript to a full review in English in a certified company. If necessary, we can present the English language edition certificate.
Thank you for the corrections to improve the manuscript
Sincerely,
Authors
Reviewer 2 Report
Comments and Suggestions for Authors
The authors have made all the changes that were suggested. I have no further comments.
Author Response
We have improved the description of methods and the way in which results are presented
Thank you for the corrections to improve the manuscript
Sincerely,